# Odorant-Binding Proteins and Chemosensory Proteins in *Spodoptera frugiperda*: From Genome-Wide Identification and Developmental Stage-Related Expression Analysis to the Perception of Host Plant Odors, Sex Pheromones, and Insecticides

**DOI:** 10.3390/ijms24065595

**Published:** 2023-03-15

**Authors:** Chen Jia, Amr Mohamed, Alberto Maria Cattaneo, Xiaohua Huang, Nemat O. Keyhani, Maiqun Gu, Liansheng Zang, Wei Zhang

**Affiliations:** 1National Key Laboratory of Green Pesticide, Guizhou University, Guiyang 550025, China; 2Key Laboratory of Green Pesticide and Agricultural Bioengineering (Ministry of Education), Guizhou University, Guiyang 550025, China; 3Department of Entomology, Faculty of Science, Cairo University, Giza 12613, Egypt; 4Division of Invertebrate Zoology, American Museum of Natural History, 200 Central Park West, New York, NY 10024, USA; 5Department of Plant Protection Biology, Swedish University of Agricultural Sciences, Box 190, Lomma—Campus Alnarp, 234 22 Lomma, Sweden; 6Department of Microbiology and Cell Science, University of Florida, Gainesville, FL 32611, USA

**Keywords:** *S. frugiperda*, developmental stages, odorant-binding proteins, chemosensory proteins, feeding behavior, sex pheromones, pesticides

## Abstract

*Spodoptera frugiperda* is a worldwide generalist pest with remarkable adaptations to environments and stresses, including developmental stage-related behavioral and physiological adaptations, such as diverse feeding preferences, mate seeking, and pesticide resistance. Insects’ odorant-binding proteins (OBPs) and chemosensory proteins (CSPs) are essential for the chemical recognition during behavioral responses or other physiological processes. The genome-wide identification and the gene expression patterns of all these identified *OBPs* and *CSPs* across developmental stage-related *S. frugiperda* have not been reported. Here, we screened for genome-wide *SfruOBPs* and *SfruCSPs*, and analyzed the gene expression patterns of *SfruOBPs* and *SfruCSPs* repertoires across all developmental stages and sexes. We found 33 *OBPs* and 22 *CSPs* in the *S. frugiperda* genome. The majority of the *SfruOBP* genes were most highly expressed in the adult male or female stages, while more *SfruCSP* genes were highly expressed in the larval or egg stages, indicating their function complementation. The gene expression patterns of *SfruOBPs* and *SfruCSPs* revealed strong correlations with their respective phylogenic trees, indicating a correlation between function and evolution. In addition, we analyzed the chemical-competitive binding of a widely expressed protein, *SfruOBP*31, to host plant odorants, sex pheromones, and insecticides. Further ligands binding assay revealed a broad functional related binding spectrum of *SfruOBP*31 to host plant odorants, sex pheromones, and insecticides, suggesting its potential function in food, mate seeking, and pesticide resistance. These results provide guidance for future research on the development of behavioral regulators of *S*. *frugiperda* or other environmentally friendly pest-control strategies.

## 1. Introduction

Insects are among the organisms with robust behavioral and physiological adaptations to complex and competitive ecosystems [1,2]. Among their various sensory capacities, insects use chemosensory perception to initiate several behavioral and physiological responses [3]. For sensing chemicals, several chemosensory neurons house specific sensilla that are located on different body parts of the insect, including the antennae, wings, mouthparts, ovipositor, legs, and other appendages [4]. In general, odors and tastant molecules diffuse through epidermal pores on the sensillar surface, and they bind to odorant-binding proteins (OBPs) or to chemosensory proteins (CSPs), also known as olfactory segment D-like proteins (OS-Ds), which are soluble proteins that are secreted at high concentrations in the sensillar lymph and are responsible for bringing odors and tastants to specific chemosensory receptors [5].

The insect’s antenna is one of the main organs on which the expression of binding proteins, like *OBPs* and *CSPs*, play important functions in chemosensation and, as a consequence, in the behavioral modulation of the insect [6]. In this and other chemosensory organs, the role of *OBPs* and *CSPs* is to deliver ligands to different chemosensory receptors, including olfactory receptors (ORs), ionotropic receptors (IRs), gustatory receptors (GRs), or sensory neuron membrane proteins (SNMPs), which are situated in the dendritic membrane of the olfactory neurons [7,8]. These, in turn, are associated with the opening of cation channels for converting the chemical signal to an electrochemical signal, which is the basis of the neuronal modulation for behavioral and other physiological responses [9]. After the activation of the chemosensory receptors, odorant degrading enzymes (ODEs) rapidly degrade the odorants [10]. However, other studies have demonstrated that the expression of odorant-binding proteins in other tissues such as the salivary glands, fat body, or midgut may be associated with alternative physiological functions in the insect [11,12,13]. Some *OBPs*, for instance, have been found to relate to innate immunity, pesticide resistance, and anti-inflammation [14,15,16]. The expression of *CSPs* has been reported in diverse insect tissues [17,18] to function in pheromone delivery, nutrient transportation, visual pigmentation, development, immunity, or pesticide resistance [19,20,21,22,23]. 

Both *OBPs* and *CSPs* are water-soluble, low-molecular-weight acidic proteins of approximately 13–17 kDa with a polypeptide length of about 120–170 amino acid residues [24,25]. *OBPs* are provided with an asset of conserved cysteine (Cys) residues [26] distributed within a hydrophobic cavity formed by at least six α-helix domains [27], which represents a conserved structural feature among the *OBPs* of most insects. In general, based on the number of conserved Cys, insect *OBPs* can be divided into 5 types: (1) classical *OBPs* with 6 conserved Cys, (2) plus-C *OBPs* with 8 conserved Cys, (3) minus-C *OBPs* with 4 conserved Cys, (4) dimer *OBPs* with 12 conserved Cys, and (5) atypical *OBPs* with 9–10 conserved Cys [28]. *OBPs* in general follow a regular pattern alternating cysteines with other amino acids (C-pattern), which is more or less conserved to facilitate the binding to ligands [29,30]. Varying among different orders, the following pattern, C1-X25-30-C2-X3-C3-X36-42-C4-X8-14-C5-X8-C6, where X is any amino acid, has been described in Lepidoptera, and a few cases have been recently reported in *Spodoptera exempta* [31].

Most insect *CSPs*, instead, have four conserved cysteine residues forming two disulfide bridges [32]. In lepidopterans, the C-pattern of *CSPs* is C1-X6-C2-X18-C3-X2-C4 [33]. Compared with *OBPs*, the conservation of *CSPs* is more distinct: insect *CSPs* maintain a high homology within species, between different orders, and across families. For example, *CSPs* are 50% to 60% homologous between *Schistocerca gregaria* and *Locusta migratoria*, and 37% to 50% homologous between Orthoptera and Lepidoptera [34]. In contrast, most insect *OBP* sequences differ from order to order and are typically less than 20% identical between genera [35]. The expression of insect *CSPs* can be within (antennae) or outside the olfactory organs, with both chemosensory (e.g., taste) and nonchemosensory roles (e.g., development) [36].

Among insects, *S. frugiperda* (J. E. Smith, 1797) (Lepidoptera: Noctuidae) is a pervasive noctuid pest in the Americas and has become invasive in Africa and Asia with a life span around 30 days at 28 °C [37]. *S. frugiperda* has many behavioral adaptations including a broad feeding range of more than 180 plant species, a strong migratory ability, a high reproductive capacity, and a strong pesticide resistance, resulting in its being a difficult target for pest-management strategies [38,39]. Interestingly, many ways of adaptation of this insect are dependent on its developmental stage. For instance, young larvae of *S. frugiperda* prefer to feed on cotton leaves, and they progressively change their preferences to fruiting structures, like squares and bolls, in older instars [40]. Older *S. frugiperda* larvae display cannibalism and stronger microbial and chemical pesticide resistance than younger larvae [41,42]. Male mate-seeking and female egg-laying site selection behaviors are common behavioral adaptations in these species by metamorphosis into adults with flying ability [43]. As a consequence, the various developmental and sex-related behavioral and physiological adaptations of this species have enabled its fast global spread and severe damage to various crops. Given the broad evidence of the diverse behavioral and physiological functions of *OBPs* and *CSPs* in insects, a genome-wide investigation and a better understanding of the phylogenetic relationships, gene structure, and the expression patterns of these proteins in *S. frugiperda* is essential.

Recently, the in-sight/in-depth view of the genome of *S. frugiperda* has been released [44], but the chemosensory proteins have previously been identified only based on transcriptomic data [45,46]. By providing a wide genomic investigation, we identified an asset of *S. frugiperda*
*OBP* and *CSP* genes to investigate their expression patterns across developmental stages and sex classes, including eggs, first to sixth instars (=larvae, in a strict sense), pupae, adult females, and males. Among the *OBPs* identified in this study, we reported *SfruOBP*31, renowned as an ortholog of *D. melanogaster* DmelOBP69a [47], being expressed in larvae and adults and binding to specific host plant volatiles, sex pheromones, and pesticides.

## 2. Results

### 2.1. Identification of *OBP* and *CSP* Genes in S. frugiperda

We identified 33 *OBP* and 22 *CSP* genes in the *S. frugiperda* sequence (Table 1 and Table 2). The *OBP* sequence multiple alignment performed with DNAMAN was 12.43%, while the *CSP* was 24.91%. All the *OBPs* we have found have conserved cysteines C3 and C4, except for *OBP18* and *OBP23*. Twenty-five *OBPs* belong to the classical *OBP* group (*SfruOBP*1–14, 16, 19–21, 24, 26–28, 30, 31, and 33), provided with six conserved cysteine residues. Among them, *SfruOBP*10, 16, 19, 20, 27, 28, and 30 miss the conserved C1, and *SfruOBP*9 misses the conserved C2, although classified as a classical *OBP* given the conservation of the other five conserved cysteines. Three of the *OBPs* we have identified belong to the minus-C *OBP* group (*SfruOBP*17, 22, and 32), provided with four conserved cysteines, missing C2 and C5. *SfruOBP*15, 25, and 29 have six conserved cysteines, and they also carry an extra cysteine downstream of C6, being classified as Plus-C *OBPs*. For *SfruOBP*18, provided with only the conserved C5, and for *SfruOBP*23, missing conserved cysteines, we cannot determine which *OBP* group they belong to, although we have identified *OBP* genes with a high homology with them in NCBI (Appendix A). Contrary to *SfruOBPs*, *SfruCSPs* are highly conserved, provided with four conserved cysteines except for *SfruCSP*1, which had no conserved cysteines; however, blast in NCBI revealed a *CSP* with a high homology, so it was identified as a putative *CSP* (Appendix A).

Gene lengths of *SfruOBPs* ranged from 413 bp (*SfruOBP*31) to 14,905 bp (*SfruOBP*23), with an average length of 3897 bp. The length of the *CSP* genes ranged from 735 bp (*SfruCSP*20) to 18,161 bp (*SfruCSP*1), with an average length of 3688 bp. The protein sequences encoded by the *OBP* genes ranged from 118 (*SfruOBP*10) to 272 (*SfruOBP*29) amino acids, with an average length of 167 amino acids. The protein sequences encoded by the *CSP* genes ranged from 107 (*SfruCSP*2) to 233 (*SfruCSP*1) amino acids, with an average length of 139 amino acids. The molecular weight of the *SfruOBP* proteins ranged from 12.69 kDa (*SfruOBP*10) to 30.59 kDa (*SfruOBP*29). The theoretical isoelectric points ranged from 4.36 (*SfruOBP*31) to 9.97 (*SfruOBP*20), respectively. The molecular weight of the *SfruCSPs* proteins ranged from 11.94 kDa (*SfruCSP*2) to 25.59 kDa (*SfruCSP*1). The theoretical isoelectric points ranged from 4.93 (*SfruCSP*22) to 10.44 (*SfruCSP*1). Most of the *OBPs*/CSPs (*SfruOBP*2-6/8/10-13/15/16/19/21/22/25/28-33, *SfruCSP*2-9/11-14/17/19/20) had a signal peptide ranging from 15 to 30 amino acids at their N-terminal, as evidence of their classification among the secretory proteins.

### 2.2. Phylogenetic Analysis of *OBP* and *CSP* in S. frugiperda

Phylogenetic analysis of the *OBPs* (Figure 1) demonstrated the existence of five clusters based on the homology among the representatives. Cluster 1 and Cluster 5 contain most of the *SfruOBPs*. Cluster 3 is the second largest cluster, containing five *SfruOBPs*, while both Clusters 2 and 4 contain three *SfruOBPs*. Most *SfruOBPs* are in the same subclade with *SlitOBPs* (Appendix A), as an indication of the close evolutionary relationship between the two species. 

Phylogenetic analysis of the *CSPs* (Figure 2) demonstrated the existence of four clusters based on the homology among the different representatives. Clusters 2 and 4 contain most of the *SfruCSPs* (12 + 7); among them, most *CSPs* of Cluster 2 belong to lepidopterans. Interestingly, Cluster 3 contains only *CSPs* from *S. invicta* (*SinvCSP*1-5/8/10/13-15/17-20). Cluster 1 contains the remaining three *SfruCSPs* (*SfruCSP*1/8/21) and four *LmigCSPs* (*LmigCSP*10/26/27/31). Like *SfruOBPs*, most *SfruCSPs* that are in the same subclade with *SlitCSPs* show a close evolutionary relationship to *S. litura *
*CSPs* (Appendix A), followed by the *CSPs* of *B. mori*.

### 2.3. Chromosomal Distribution of *OBP* and *CSP* Genes

The chromosomal location map of *OBP* and *CSP* genes was plotted with tbtools based on the information of the genome annotation file of *S. frugiperda* (Figure 3). The identified *OBP* genes of *S. frugiperda* are located on 10 different chromosomes and on contig ctg319_4 (which is a different contig not yet defined, belonging to a specific chromosome), and the majority of the *SfruOBPs* are distributed in clusters on chromosomes. For instance, nine genes from Cluster 1 (*SfruOBP*5/6/11/14/21/25/26/29/33) are located in proximity on chromosome 10, while the remaining two genes of Cluster 1 are instead distributed on chromosome 20 (*SfruOBP*10) and chromosome 22 (*SfruOBP*19). One *SfruOBP* gene from Cluster 2 is located on chromosome 2 (*SfruOBP*23), and two *SfruOBP* genes from the same Cluster are located on chromosome 22 (*SfruOBP*17/18). The *SfruOBP* genes from Cluster 3 are distributed dispersedly on chromosome 2 (*SfruOBP*8), chromosome 10 (*SfruOBP*7), chromosome 14 (*SfruOBP*31), and chromosome 25 (*SfruOBP*22/32), respectively. Two *SfruOBPs* (*SfruOBP*1/20) from Cluster 4 are located on chromosome 15 and another one (*SfruOBP*27) is located on ctg319_4. Six *SfruOBPs* from Cluster 5 (*SfruOBP*2/3/4/12/13/15) are located on the adjacent segments of chromosome 8 as well as another *OBP* gene (*SfruOBP*28), which is located on the same chromosome but at approximately 3 Mbp distance from the latter. Other *OBP* genes of Cluster 5 (*SfruOBP*16/30) are located on chromosome 3 (*SfruOBP*24), chromosome 12 (*SfruOBP*9), and chromosome 15 (*SfruOBP*16/30). 

Contrary to *SfruOBPs*, the distribution of *SfruCSPs* seems to be more centralized, having identified loci on only two chromosomes: two genes of Cluster 2 (*SfruCSP*2/20) are located on adjacent loci of chromosome 19, while the remaining 21 genes of Clusters 1, 2, 3, and 4 are all located on chromosome 8 (Figure 4). On this chromosome, most of the genes fall in proximal loci, with the exception of *SfruCSP*1 and *SfruCSP*12, located at 3.5 Mbp and 6.6 Mbp, respectively, on the chromosome.

### 2.4. Intron/Exon Organization of *OBP* and *CSP* Genes of S. frugiperda

The exon/intron structure varies greatly within *SfruOBP* (Figure 5) and SfruCSP (Figure 6) genes from different clusters. For example, *SfruOBP*2/3/4/9//12/13/28/30 from Cluster 5 have three exons each, while *SfruOBP*24 has five exons. *SfruOBP*29 (Cluster 1) contains eight exons, which is the highest number of exons. However, two genes contain only one big exon (*SfruOBP*27/31). Members of the *SfruCSPs* have fewer exons than those of the *SfruOBPs*. The majority of the *SfruCSP* genes have only two exons, except for *SfruCSP*2/10/16/18/20 with three exons, *SfruCSP*22 with four exons, and *SfruCSP*1 with five exons. When compared, the exon and gene lengths of the *SfruCSP* members differed from those of the *SfruOBP* members. 

### 2.5. Motif Analysis of *OBPs* and *CSPs* of S. frugiperda

MEME analysis revealed that there are six different motif structures in the polypeptide sequences of *SfruOBPs* (Figure 7) and five different motif structures in the polypeptide sequences of *SfruCSPs* (Figure 8). Half of the *OBPs* from Cluster 1 present the 6-1-2 motif pattern (*SfruOBP*5/6/25/26/33), and the other half present a different organization of the motif pattern (*SfruOBP*11/14/19/21/29). The three *SfruOBPs* from Cluster 2 present only motif 1 (*SfruOBP*17/18/23) as well as nine other *SfruOBPs* (*SfruOBP*1/7/9/10/19/20/23/24/28). Three *SfruOBPs* from Cluster 3 present the 6-1 motif pattern (*SfruOBP*8/31/32) that is present also for *SfruOBP*27 from Cluster 4, while *SfruOBP*22 presents the 1-6 motif pattern and *SfruOBP*7 presents only one motif (motif1). *OBPs* from Cluster 5 (*SfruOBP*2/3/4/12/13/15) present the 4-1-5-3 motif pattern.

The majority of the *SfruCSPs* present a pattern of 4-3-1-2 motifs (Figure 8). These included *SfruCSP*8 from Cluster 1, almost of the *SfruCSPs* from Cluster 2, as well as all the *SfruCSPs* from Cluster 4. The remaining five *SfruCSPs* showed four distinct motif patterns (*SfruCSP*1/2/6/20/21). Among the latter, two *SfruCSPs* from Cluster 2 present the 4-1-5 motif pattern (*SfruCSP*2/20), *SfruCSP*6 presents only motif 4, *SfruCSP*21 presents the 4-1 motif pattern, and *SfruCSP*1 presents only motif 2.

### 2.6. Gene Expression Analysis of *OBPs* and *CSPs* of S. frugiperda

Our transcriptomic dataset was used to create a heatmap revealing expression levels of *SfruOBPs* and *SfruCSPs* across the developmental stages of egg, first to sixth instars (identified by the size of the larvae), and pupae, as well as adult males and females (Figure 9). Transcriptomic analysis revealed that the expression of almost all of the genes of Cluster 1 was higher in the first instar and decreased in the second to sixth instars and the pupal stage, but recovered to a higher level in the adult stage. However, for *SfruOBP*19, expression appeared stable in both the larval and adult stages, and *SfruOBP*11 showed high expression only in the first instar and maintained a low expression for all the other developmental stages. Two genes in Cluster 2 (*SfruOBP*17/18) were stably expressed at all stages except eggs and the first instar, while *SfruOBP*23 was expressed at instar one through five and showed a decreasing trend and absent to low expression from the pupal to adult stages. The expression of *SfruOBP*7 and *SfruOBP*31 from Cluster 3 was higher in the first instar. Three other genes (*SfruOBP*8/22/32) were highly expressed in the pupae and adults. Two genes of Cluster 4 (*SfruOBP*20/27) and two genes of Cluster 5 (*SfruOBP*16/30) were highly expressed in both the male and female adults. *SfruOBP*20 was also highly expressed in the first and fourth instars, and *SfruOBP*1 in Cluster4 was highly expressed in the first to third instars. Some of the genes from Cluster 5 were highly expressed in the egg stage, while all genes from this cluster were highly expressed in adults. Genes of Cluster 5 were also moderately to highly expressed in several earlier or later larval stages: *SfruOBP*3/13/15 were highly expressed in earlier larval stages (instars 1–3), *SfruOBP*4/12 were highly expressed in later larval stages (instars 4–5), while *SfruOBP*2 was expressed in all instars, and *SfruOBP*9/24/28 were also expressed in most stages of the larva. It is worth mentioning that *SfruOBP*9 was highly expressed in males but low in females.

The majority of the *SfruCSPs* showed a more complex expression tendency across the developmental stages and sexes (Figure 10) when compared with *SfruOBPs*. In Cluster 1, *SfruCSP*21 gradually decreased in expression level from eggs to the sixth instar but was highly expressed in pupae and adults, *SfruCSP*8 was highly expressed only in adults, and *SfruCSP*1, from a low to absent expression in eggs, maintained a moderate expression in all developmental stages. All the genes from Cluster 2 were moderately or highly expressed from the egg to pupal stages. However, *SfruCSP*11/16 showed low expression in the sixth instar, *SfruCSP*17 showed low expression in the fourth to sixth instars and in the egg stage, *SfruCSP*5 showed low expression in the fourth instar, and *SfruCSP*4 showed a moderate expression for all larval stages except for the sixth instar. Two genes showed a relatively low expression in pupae (*SfruCSP*6/13), and only three genes of this cluster exhibited high expression levels in adult males and females (*SfruCSP*4/6/17). Within the same cluster, *SfruCSP*2, after its high expression in eggs, gradually increased expression levels from the first instar to the pupal stage, dropping to low levels in adults. Within the same cluster, *SfruCSP*6 showed an overall high expression except for the first instar and the pupae, while *SfruCSP*20 showed a sort of *off-on* expression pattern throughout all the developmental stages of the insect, from eggs to adults. Except for *SfruCSP*15/22, all genes from Cluster 4 were highly expressed from the first instar to the fifth instar, where *SfruCSP*18 showed a reduced expression. Expression of the genes from Cluster 4 seems to be consistent in both adult males and females. Interestingly, there was low expression of *SfruCSP*7/9/14 in pupae and of *Sfru*3/7/18 in the sixth instar. *SfruCSP*22 was highly expressed in adult females and showed a lower expression across the other developmental stages.

Among the examined genes, the expression patterns of *SfruOBP*32 (Appendix A), *SfruCSP*3 (Appendix A), *SfruCSP*19 (Appendix A), and *SfruCSP*20 (Appendix A) were similar across all developmental stages comparing the qRT-PCR data with the transcriptomic data. The expression of *SfruOBP*24 (Appendix A) in four stages (fifth instar, sixth instar, pupae, and female adult) was consistent between the qRT-PCR and transcriptomic data, but the qRT-PCR analysis also revealed moderate expression of *SfruOBP*24 at the fourth instar, third instar, first instar, and egg stages and high expression of *SfruOBP*24 at the second instar and the male adult stages. The qRT-PCR results revealed an adult male biased expression pattern of *SfruOBP*22. The expression of five stages (L2, L3, L4, L6, and AM) of *SfruOBP*17 was also consistent between the qRT-PCR and transcriptomic data (Appendix A). 

### 2.7. Binding of *SfruOBP*31 to Host Volatiles, Pheromones, and Pesticides

*SfruOBP*31 showed binding capacity to 1-NPN with a Ki of 22.4 μM, indicating 1-NPN as a proper fluorescence reporter (Figure 11A). A chemical-competitive binding assay unveiled *SfruOBP*31 binding to linalool, cis-3-hexenyl acetate, 1-nonanol, and decanal, with the following Ki values: 4.04, 6.38, 14.17, and 19.56 μM, respectively. Among sex pheromones, *SfruOBP*31 binds (Z)-11-hexadecenyl acetate and (Z)-9-tetradecenyl acetate with Ki values of 5.40 and 5.02 μM, respectively, but not to (Z)-7-dodecenyl acetate or (Z)-9-dodecenyl acetate. Among insecticides, we reported *SfruOBP*31 binding to lambda-cyhalothrin (Ki = 2.78 μM) and chlorfenapyr (Ki = 8.1), but a lack of binding to emamectin benzoate (Ki = 27.55 μM), lufenuron (Ki = 37.58 μM), and chlorantraniliprole (Ki= 22.28 μM).

## 3. Discussion

*S. frugiperda* is an important and invasive agricultural pest whose destructiveness occurs across life stages, behavior patterns, and physiological changes [48]. In this paper, by performing a genome-wide screening of *S. frugiperda,* we have unveiled its asset of *OBPs* and *CSPs*. Based on a phylogenetic analysis of *SfruOBPs* and *SfruCSPs*, we compared the gene structures, polypeptide sequence motif patterns, and patterns of gene expression across all developmental stages of the insect, from eggs to adults. We chose one of the highest expressed binding proteins of *S. frugiperda*, *SfruOBP*31, that we heterologously expressed and purified from transgenic *E. coli* (Appendix A) to assess chemical-competitive binding assays demonstrating binding to several ligands among odors emitted from the host plant *Z. mays*, sex pheromones, and some of the main pesticides for this insect.

In the last decades, most of the investigation of lepidopterans targeted their chemoreceptors [49,50,51,52]. Among various reports, some provided in-depth analysis of specific *OBPs* [53,54], *CSPs* [18,55], or both [30,56]. However, to our knowledge, this is among the few [57], or may be the sole study in lepidopterans, in which a systematic investigation of soluble proteins merges with a variegate analysis including genomic loci, gene and polypeptide structures, expression patterns across every developmental stage, and functional evidence of semiochemical/pesticide binding. 

By mediating the ligand transport through the sensillar endolymph to the activation of chemosensory transmembrane proteins [7,8], *OBPs* and *CSPs* play a crucial role in insect behavioral and physiological adaptation, including food seeking, reproduction, and pesticide resistance [58]. 

Belonging to two multigene families, *OBPs* and *CSPs* generally differ in numbers and functionalities across insect species [21]. To our knowledge, this is among the first studies identifying *OBPs* and *CSPs* with the in-sight/in-depth view of the genome of *S. frugiperda*, wherein we demonstrate the existence of 33 genes for *OBPs* and 22 genes for *CSPs*. Our findings are consistent with a previous study conducted on *S. frugiperda* where 36 *OBPs* and 21 *CSPs* were identified [45], with 22 *SfruOBPs* and 19 *SfruCSPs* identical with our study (Appendix A) and comparable with the number of the respective genes for *S. exigua* (34 *OBPs*, 20 *CSPs*) [59]. 

Taking all binding proteins together, *S. frugiperda* expresses fewer *OBPs* and *CSPs* than *B. mori* (44 *OBPs*, 20 *CSPs*) [33,57] and *S. litura* (45 *OBPs*, 23 *CSPs*) [60]. The *SfruOBPs* were classified according to Hekmat-Scafe et al. [61]. *SfruOBPs* with Cys at only six conserved sites were classified as classical *OBP*, while *SfruOBPs* in the absence of C2 and C5 were classified as minus-C *OBP*. *SfruOBPs* with more than six conserved Cys were classified as plus-C *OBPs*. Among the 33 identified *SfruOBP* genes, 22 belonged to the classical group; only three *SfruOBPs*, which lacked C2 and C5, belonged to the minus-C group (*SfruOBP*22/32); and three genes belonged to the plus-C group (*SfruOBP*15/25/29). This is comparable to *S. litura*, in which only classical, minus-C, and plus-C *OBPs* were identified [62,63]. However, one dimer *OBP* was annotated in *Danaus plexippus* [53]. The majority of the *SfruCSPs* contained four conserved Cys, and only three *CSPs* contained three conserved Cys. 

Our phylogenetic analysis demonstrates evolution relationships of *OBPs* and *CSPs* between *S. frugiperda* and other species of Lepidoptera; specifically, between *S. frugiperda* and *S. litura* (Figure 1). According to the phylogenetic tree, we found five clusters of *SfruOBPs*, among which some *OBPs* from the same cluster have similar gene structures and expression patterns, while some *OBPs* from different clusters present different gene structures and expression patterns. The phylogenetic tree of *CSPs* was divided into four clusters, of which only three out of four contain *CSPs* from *S. frugiperda*. Among these, Cluster 2 and Cluster 4 contain most of the *SfruCSPs*, as a possible indication of their similarity in evolution and possibly as evidence of differences in foraging behavior or adaptation to the environment between *S. frugiperda* and *S. litura* [64]. 

Chromosomal location, gene structure, and protein motif analysis may help unveil potentials from gene evolution such as duplication, reversal, or skipping [62,65] to functional conservation. In general, about half of the *SfruOBPs* are distributed on multiple chromosomes, whereas the *SfruCSPs* are distributed on only two chromosomes, and most of them are clustered on adjacent loci of one chromosome, which may be due to the high conservation of the gene family in *S. frugiperda*. In addition, the *SfruOBP* genes display divergent patterns of intron/exon organization (Figure 5), which, although speculative, may widen a potential for alternative splicing, as known for a different class of receptors in lepidopterans [66,67], and motif patterns that are only partially conserved among the representatives belonging to the same clusters (Figure 7). The evolution of the *OBP* gene family is suggested to follow the birth-and-death model through the pseudogenization or the functional divergence of the duplicate gene during duplication [68,69]. However, the adjacent genes on chromosomes could be involved in analogous functions. For instance, 10 of 24 *OBPs* located in the social chromosome of *S. invicta* may participate in the behavioral modulation between the monogyne colony and the polygyne colony [70]. Unlike *SfruOBPs*, the majority of *SfruCSPs* from four clusters are instead distributed in proximity on the same chromosome (chromosome 8), evidence of their possible origins from gene duplication. In addition, several *SfruCSP* genes display similar intron/exon organization (Figure 6), and analysis of their expressed amino acid sequences demonstrates the existence of an identical motif pattern, 4-3-1-2 (Figure 8). *SfruCSP*2 and *SfruCSP*20 are located in a separate chromosome, suggesting their divergent functions. As a homolog to *SfruCSP*20, *SlitCSP3* was revealed to function in plant defensive metabolites recognition [71]. The functions of the chromosomal location close and far *SfruOBPs*/*SfruCSPs* are worthy of further study to verify their function evolution.

Transcriptomic analysis unveiled that six genes of *OBPs* of the *SfruOBP*-Cluster 5 (*SfruOBP*2/3/4/12/13/15), five genes of *OBPs* of the *SfruOBP*-Cluster 3 (*SfruOBP*7/8/22/31/32), two genes of *OBPs* of the *SfruOBP*-Cluster 4 (*SfruOBP*20/27), two genes of *OBPs* of the *SfruOBP*-Cluster 5 (*SfruOBP*16/30), and two genes of *OBPs* of the *SfruOBP*-Cluster 1 (*SfruOBP*10/19) showed high expression in adults, suggesting that they have important functions in the adult stage (Figure 9). RNA interference of highly expressed *OBP*1 in the adult antennae of *Culex quinquefasciatus* led to significantly reduced electrophysiological responses to egg-laying attractants, indicating that *CquiOBP*1 may mediate spawning behavior [72]. Although speculative, this may suggest potential olfactory functions of the *OBPs* that are highly expressed in adults’ *SfruOBPs* and their involvement in behaviors such as spawning but also courtship or reproduction. The majority of *SfruOBPs* showed low expression at the sixth instar, whereas we demonstrated higher expression for representatives of Cluster 2. We assume that the lower expression of *SfruOBPs* at this larval stage may be associated with delayed feeding as a prepupal physiological feature from older larvae. As expected, most *SfruOBPs* showed low or moderate expression in the egg stage, with the sole exception of *SfruOBP*12, which may be associated with some sort of role in embryo development like in *Galeruca daurica,* where *GdauOBP*28 also showed an exclusively high expression in the egg stage [73]. Contrary to our expectations, the expression of several, if not the majority, of *SfruOBPs* was higher in the earlier instars than in the older ones. Among these, *SfruOBP*11 was the *OBP* with the highest expression in the first instar, which, hypothetically, may suggest its role in conspecific recognition or defensive mechanisms at the larval stage. Indeed, studies have shown the role of *OBP* in binding chemical ligands and regulating behavior, like the larval-specific *OBP* of *S. exigua,* SexiOBP13, which showed a high binding ability to the sex pheromone, Z9, and E12-14:Ac [74]. In the pea aphid *Acyrthosiphon pisum*, *ApisOBP*3 is expressed in old larvae and in apterous adults [75] to function in short-term defensive responses like feeding cessation and dropping from host plants in a phase of the behavioral response to the alarm pheromone (E)-β-farnesene [76]. In addition, *OBPs* from the same clusters have similar gene structures, which can help us to classify *OBPs* that may have similar functions for further study. For example, *SfruOBP*2/3/4/12/13/15 from Cluster 5 present the 4-1-5-3 motif pattern (Figure 7), and all contain loci clustered on the same chromosome, organizing three exons, as an indication of their similarity in function.

Analyzing *CSP* expression, we demonstrated that most of the *CSPs* we identified were expressed throughout the whole life cycle of *S. frugiperda*, especially among the various instars (Figure 10). Compared with *OBPs*, *CSPs* were more abundantly expressed at different developmental stages, which may be associated with their broader physiological functions and tissue distribution than *OBPs*, being involved in leg regeneration [77] and sucking [20]. The expression of some of the genes from Cluster 2 (*SfruCSP*2/5/10/11/12/13/16/19/20) was lower in the adult stages, but in comparison, it was higher in the later larval stages. These genes may be related to the developmental-stage-dependent behavioral or physiological adaptations in older larvae. Genes of Cluster 4 (*SfruCSP*15/22) and *SfruCSP*8 were highly expressed during adulthood, indicating a function in behaviors such as courtship, oviposition, or reproduction [78]. For example, the adult expression with a female bias of *SfruCSP*22 suggests this gene’s involvement in ligand binding to the neuronal modulation for behaviors forming the basis of choosing suitable sites for egg-laying or other female-specific activities [79]. Several *SfruCSPs* were expressed in the egg stage, among which the highest expression belonged to the *CSPs* from Cluster 2 (*SfruCSP*2/6/11/13). Studies have shown that *CSP* has odor recognition functions, such as the *AlinCSP*1-3 in the alfalfa plant bug, *Adelphocoris lineolatus*, binding to several host-related ligands including (Z)-3-hexen-1-ol, (E)-2-hexen-1-al, and valeraldehyde, as a possible involvement of these *CSPs* in host recognition [80]. Beyond chemosensation, these *CSPs* may have functional properties forming the basis of embryonic development, physiologically or morphologically [81]. For example, the honey bee gene *AmelCSP*5 is associated with the formation of the embryonic covering membrane [82].

Previous reports on *Spodoptera*
*OBPs* provides a blueprint for the prediction of SlitOBP ligands based on the interaction of phylogeny and chemical structure. For example, larval *SlitGOBP*2 of *S. litura* have been reported to function in sex pheromone recognition [83]. In the same species, *OBP*11 displays strong binding with sex pheromones [84] and *CSP*8 can bind with the oviposition deterrent chemical rhodojaponin III [85]. *CSP*18 binds directly to chlorpyrifos/fipronil and *CSP*6 to chlorpyrifos, emamectin benzoate, and fipronil [86]. In addition, three *CSPs* of *S. litura* (*SlitCSP*11, 3, and 8) that are highly expressed in the midgut of the insect were shown to bind to host plant chemicals, and their expression levels varies depending on the host plants [80]. In this study, we verified the function of a widely expressed *OBP*, *SfruOBP*31, using a chemical-competitive binding assay based on measurements of 1-NPN-associated fluorescence (Figure 11). The reason we chose *SfruOBP*31 for binding assays is its high expression in the first and fifth larval instars and in adults (Figure 9). Indeed, these developmental stages were reported to be involved in some remarkable physiological and behavioral adaptions, including the dispersing behavior of the first instar [87], higher pesticide resistance in elder larvae [88], and the egg laying/mate seeking behavior of adults [89,90]. Our phylogenetic analysis clustered *SfruOBP*31 in Cluster 3, where, according to our transcriptomic analysis (Figure 9), it is represented by *OBPs* of *S. frugiperda* with the high expression patterns in the first instar or in the fifth instar and adults, suggesting its potential multiple functions, including the dispersing behavior in the first instar, higher pesticide resistance in elder instars, and the egg laying/mate seeking behavior of adults. Among other developmental stages, *SfruOBP*31 was also low to moderately expressed. In addition, *SfruOBP*31 is the ortholog of *DmelOBP*69a in *Drosophila*, sharing 81% query cover and 28.57% identity, and is involved in social interactions by the detection of contact sex pheromones [47]. Among lepidopterans, sex pheromones are key to mating behavior in adults, and previous studies demonstrated that the sex pheromone mixture of *S. frugiperda* is composed of (Z)-9-tetradecen-1-yl acetate (Z9-14:Ac), (Z)-7-dodecen-1-yl acetate (Z7-12:Ac), (Z)-9-dodecen-1-yl acetate (Z9-12:Ac), and (Z)-11-hexadecen-1-yl acetate (Z11-16:Ac) in an 81:0.5:0.5:18 ratio [91]. Here, we tested the binding ability of *SfruOBP*31 to these four sex pheromones. Competitive binding assays with 1-NPN demonstrated that *SfruOBP*31 selectively binds with two sex pheromones ((Z)-11-hexadecenyl acetate and (Z)-9-tetradecenyl acetate); this may be due to the high content of these two compounds in the sex pheromone mixture of *S. frugiperda*. Binding of a larval-specific *OBP* to odorant pheromones is not surprising in the genus *Spodoptera.* Indeed, a larval-specific *OBP* of *S. exigua*, SexiOBP13, shows strong binding capacity to the female sex pheromone Z9,E12–14:Ac that can induce the attraction behavior in larvae [74]. However, evidence of the expression of this *OBP* also in adults with a similar level between males and females (Figure 9) may suggest involvement of this binding protein to different ecological mechanisms. 

The insect olfactory system adapts to the environment by detecting specific chemical volatiles [92]. *S. frugiperda* can locate their host plants through plant odors. Previous studies have revealed that many plant odors attract *S. frugiperda*, such as linalool [93]. Therefore, we measured the binding capacity of *SfruOBP*31 to five host plant volatiles (Figure 11). Our findings demonstrate that our host plant odorants linalool, cis-3-hexenyl acetate, 1-nonanol, and decanal associate fluorescent decrement in competitive assays conducted with 1-NPN with the following Ki values: 4.04, 6.38, 14.17, and 19.56 μM, respectively. These findings provide evidence of the ability of *SfruOBP*31 to recognize odors emitted from host plants while the neonate larvae are emerging from the egg mass. 

OBP has been shown to bind to insecticide components and to be involved in the increment of resistance. For instance, GOBP2 is responsible for chlorpyrifos tolerance in *S. litura* [94]. In other insects like *Nilaparvata lugens*, *OBP*3 is involved in the resistance to nitenpyram and sulfoxaflor [15]. In our binding trials we also tested the binding capacity of *SfruOBP*31 to five pesticides commonly used in the control of *S. frugiperda*, demonstrating that *SfruOBP*31 had the strongest binding capacity to lambda-cyhalothrin (Ki = 2.78 μM) and chlorfenapyr (Ki = 8.1 μM). Evidence of binding to pheromones, plant odorants, and pesticides suggests multiple functions for *SfruOBP*31 in host plant recognition and pesticide resistance, and as demonstrated for its ortholog in *Drosophila, DmelOBP69a*, involved in social interaction based on binding with sex pheromones [47]. 

In conclusion, we performed a systematic identification of *SfruOBPs* and *SfruCSPs* based on the in-sight/in-depth view of the genome and analyzed their expression patterns across all different developmental stages and adult males and females. The different expression patterns of some *SfruOBPs* and *SfruCSPs* have developmental stage specificity or sex biases, suggesting the potential functions of these genes. These results lay the groundwork for future investigation of their in vivo functions with methods such as CRISPR/Cas9 or modified RNAi [95,96]. The in vitro chemical competitive binding assay of *SfruOBP*31 further confirmed the involvement of some *OBPs* or *CSPs* in multiple functions such as host plant plus pheromone binding or pesticide resistance. Our study opens investigations on behavioral and physiological adaptation mechanisms of the Fall armyworm *S. frugiperda* for the final target to develop alternative control strategies to interfere with the behavior or other physiological functions of this pest.

## 4. Materials and Methods

### 4.1. Gene Identification

The amino acid sequences of *OBPs* and *CSPs* from *B. mori, S. invicta*, *D. melanogaster*, *L. migratoria,* and *S. litura* were selected as templates to search against the genome of *S. frugiperda.* Homology searching was conducted using the tBLASTn programs with the E-value cut-off of 0.0001 to retrieve alternative protein sequences of *S. frugiperda*
*OBPs* and *CSPs*. HMMER software (version 3.0) [97] was used in search of unique domains of the corresponding gene family in the alternative protein sequences. Genes that contain a conserved PBP_GOBP domain for the *OBP* genes (accession number pfam01395) and a conserved OS-D domain for the *CSP* genes (accession number pfam03392) were maintained. These genes were then manually verified based on the conserved Cys residue characteristics of lepidopterans’ *OBP* genes (C1-X_25-30_-C2-X_3_-C3-X_36-42_-C4-X_8-14_-C5-X_8_-C6) and *CSP* genes (C1-X_6_-C2-X_18_-C3-X_2_-C4) [30]. After filtering the sequences without characteristic domains, the protein sequences belonging to this gene family were kept for further analysis (Appendix A).

### 4.2. Sequence Alignment and Phylogenetic Tree Construction

The phylogenetic tree of *S. frugiperda*, *L. migratoria*, *D. melanogaster*, *S. litura*, *B. mori*, and *S. invicta*
*OBP* and *CSP* genes was constructed to reveal the evolutionary relationship of *SfruOBPs* and *SfruCSPs* among species. Amino acid motifs were identified using MEME [98]. The amino acid multiple sequence alignment (MSA) was made using the muscle method with MEGA 11.0, from which we estimated the best-fitting model of amino acid substitution [99]. Phylogenetic trees were built using MEGA 11.0, where branch lengths were optimized, and branch supports were calculated using bootstrapping with 1000 replicates of neighbor joining (NJ). To facilitate future research, the *OBP* and *CSP* genes of *S. frugiperda* were named based on the homology of genes from other species found using blastx in NCBI.

### 4.3. Chromosomal Distribution, Gene Structure, and Motif Analysis of *OBPs* and *CSPs*

Chromosomal location information of *SfruOBP* and *SfruCSP* genes was downloaded from genome data [44]. We visualized the data using TBtools [100], including the location and length of the genes on the chromosome and the visualization of the exon-intron structure of the *SfruOBP* and *SfruCSP* genes. To discover the motif pattern of *SfruOBPs* and *SfruCSPs*, 33 *OBPs* and 22 *CSPs* were submitted to the MEME (version 5.5.1) online tool (https://meme-suite.org/meme/, accessed on 9 December 2022) server and a Zero-or-One Occurrence per Sequence (zoops) distribution pattern was adopted.

### 4.4. Samples Preparation, RNA Extraction, and RNA-seq

*S. frugiperda* were reared individually in plastic boxes with fresh corn leaves. Insects were maintained at 28 ± 3 °C under 70–75% relative humidity and a photoperiod of 16 h light, 8 h dark cycles. Adults of *S. frugiperda* were reared in 30 cm × 40 cm mesh cages supplied with fresh 10% honey water and corn seedlings for egg laying. Eggs were collected in 24 h after they were laid. Synchronized 1st-to-6th instars were collected according to their size. Three-day pupae and unmated adults were also collected. For each replication, RNA was extracted from 300 eggs and the whole body of 200, 100, 50, 10, 5, and 5 larvae at their 1st, 2nd, 3rd, 4th, 5th, and 6th stages, respectively; 5 pupae; 5 adult females; and 5 adult males. All stages were collected with three replications.

Trizol Reagent (Invitrogen, Waltham, MA, USA) was used for total RNA extraction according to the manufacturer’s instructions. Genome DNA was extracted using Turbo DNase (Thermofifisher, Waltham, MA, USA). Gel extraction was performed, and a Nanodrop ND-1000 spectrophotometer (LabTech, Hopkinton, MA, USA) was used to assess the quality and quantity of the RNA samples. RNA libraries were constructed and sequenced as described previously [101]. RNA-seq was performed by a commercial company (Lianchuan, Hangzhou, China) with a next-generation sequencing platform (Illumina NovaseqTM 6000). Differentially expressed genes across all developmental stages were analyzed using the DESeq R package (1.10.1).

### 4.5. q-RT-PCR Verification of Selected SfruOBPs and SfruCSPs

Beacon Designer Software (Palo Alto, CA, USA) was used to design primers (Appendix A), with *RPL32* (Ribosomal Protein L32) as the reference gene. Q-RT-PCR experiments were performed in accordance with the Minimum Information Required for Publication of Quantitative Real-Time PCR guidelines. A Premix Ex TaqTM II (Tli RNaseH Plus) Kit (Takara, Shiga, Japan) was used as a reagent for qRT-PCR analysis. The reaction conditions were set as follows: 95 °C for 3 min, 40 cycles of 95 °C for 15 s, and 60 °C for 30 s. Reactions were carried out in triplicate, followed by dissociation in the iCycler iQ™ Real Time PCR Detection System (Bio-Rad, Hercules, CA, USA). All developmental stages and sexes were performed with three separate biological replicates. The data were analyzed according to 2^−ΔΔCT^ [ΔΔCt = ΔCt (test) − ΔCt (calibrator)]. Four *SfruOBPs* (32/24/22/17) and three *SfruCSPs* (3/19/20) were selected for q-RT-PCR assay as representatives that are specifically expressed at different development stages.

### 4.6. Heterologous Expression and Purification of *SfruOBP*31 in Bacteria

We expressed *SfruOBP*31 from *S. frugiperda* in bacteria using a prokaryotic expression system. The predicted signal peptide was removed to produce a properly folded protein. The ORF (Open Reading Frame) of *SfruOBP*31 was cloned into a pET28a vector (Novagen, Darmstadt, Germany) and transformed into an *Escherichia coli* BL21 strain. Individual positive colonies were incubated until the OD (600 nm) reached 0.6-0.8. Protein expression was induced with 0.5 mM IPTG (isopropyl-β-D-1-thiogalactopyranoside) at 16 °C with overnight shaking at 220 rpm/min. Cells were harvested using centrifugation at 3000× *g* and resuspended in HEPES buffer (10 mM HEPES, 100 mM NaCl, pH 7.5). The supernatant and pellet were then separated after sonication and centrifugation at 14,000× *g* and 4 °C for 30 min. SDS-PAGE was used to demonstrate that the recombinant protein was soluble. The solution was next applied to his-trap affinity columns (Cobalt Chelating Resin, GBiosciences, St. Louis, MO, USA). Bound protein was eluted with HEPES buffer containing progressively increasing amounts of imidazole from 50 mM to 500 mM. The fractions with the recombinant protein were pooled and dialyzed three times against 3 L of HEPES buffer at 4 °C overnight after electrophoretic analysis. To confirm the size of the recombinant protein, his-tag antibody (Thermofisher, Waltham, MA, USA) was used to perform Western blot. The recombinant protein was stored at −80 °C until use.

### 4.7. Chemical Competitive Binding Assays

To investigate the ligand binding ability of *SfruOBP*31, we used a fluorescence competitive binding assay using the fluorescent probe N-phenyl-1-naphthylamine (1-NPN) [102]. Binding experiments were performed on a microplate reader using a Greiner 96 Black Flat Bottom. Among the ligands, we tested five host volatiles from the host plant *Zea mays*: decanal (97% purity, Shanghai Acmec Biochemical Co., Ltd., Shanghai, China), linalool (98% purity, Shanghai Acmec Biochemical Co., Ltd.), leaf alcohol (98% purity, Shanghai Acmec Biochemical Co., Ltd.), 1-nonanol (98% purity, Shanghai Acmec Biochemical Co., Ltd.), and *cis*-3-hexenyl acetate (98% purity, Shanghai Acmec Biochemical Co., Ltd.); four pheromones: (Z)-9-tetradecenyl acetate (93% purity, Shenzhen Regent Biochemical Technology Co., Ltd., Shenzhen, China), (Z)-11-hexadecenyl acetate (98% purity, Shanghai Bidd Medical Technology Co., Ltd., Shanghai, China), (Z)-7-dodecenyl acetate (90% purity, Shanghai Bidd Medical Technology Co., Ltd.), and (Z)-9-dodecenyl acetate (90% purity, Shanghai Bidd Medical Technology Co., Ltd.); and five pesticides: emamectin benzoate (90% purity, Energy Chemical), lufenuron (97.4% purity, Shanghai Yuanye Biotechnology Co., Ltd., Shanghai, China), chlorfenapyr (98% purity, Shanghai Yuanye Biotechnology Co., Ltd.), chlorantraniliprole (95% purity, Shanghai Yuanye Biotechnology Co., Ltd.), and lambda-cyhalothrin (96% purity, Guangdong Liwei Chemical Co., Ltd., Maoming, China). All chemicals used for testing and the fluorescent probe 1-NPN were dissolved in GC purify-grade methanol in a 400 µM solution. The 2.0 µM protein solution (in 10 mM HEPES and 100 mM NaCl buffer, pH 7.5) was titrated with 400 µM 1-NPN to a final concentration of 2–20 µM. The binding constant of the protein to 1-NPN was analyzed by assessing the continuous change of fluorescence value. Compounds of 2–20 µM concentration were added to the mixture of 2.0 µM 1-NPN and 2.0 µM protein or 2.0 µM 1-NPN alone to determine the binding rate of the ligands to the protein and the value of the fluorescence. The final fluorescence value was calculated as the fluorescence value after the addition of 1-NPN and protein with the competing ligand minus the fluorescence value after the addition of 1-NPN with the competing ligand only.

The dissociation constants (K_1-NPN_) of *SfruOBP*31 were calculated using Graphpad Prism 8 software. The dissociation constant (K_i_) of 1-NPN by competing ligands was calculated with the following formula using IC50 value: K_i_ = [IC50]/(1 + [1-NPN]/K_1-NPN_). Here, IC50 is the concentration of competing ligands when the fluorescence intensity reaches half of the initial fluorescence intensity of 1-NPN, and [1-NPN] indicates the concentration of 1-NPN [16]. A threshold of K_i_ < 20 µM was recognized as binding.

## Figures and Tables

**Figure 1 ijms-24-05595-f001:**
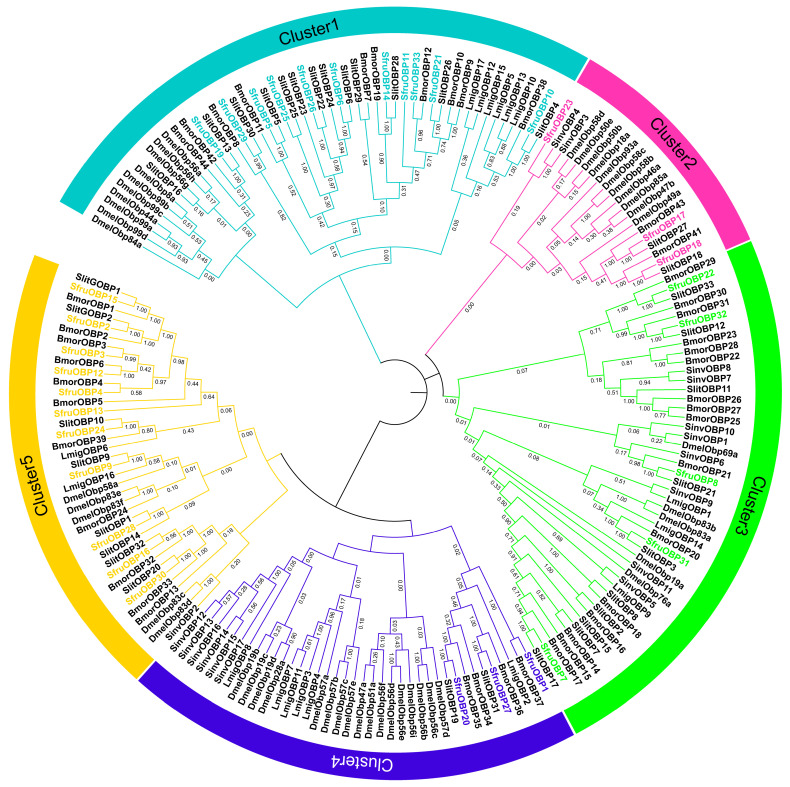
Phylogenetic analysis of *OBP* representatives from six species. The prefixes Sfru, Slit, Bmor, Lmig, Dmel, and Sinv denote *OBP* proteins from *S. frugiperda, S. litura, B. mori*, *L. migratoria, D. melanogaster,* and *S. invicta*, respectively. The *SfruOBP* genes are binned into various clusters based on the homology among *OBP* representatives.

**Figure 2 ijms-24-05595-f002:**
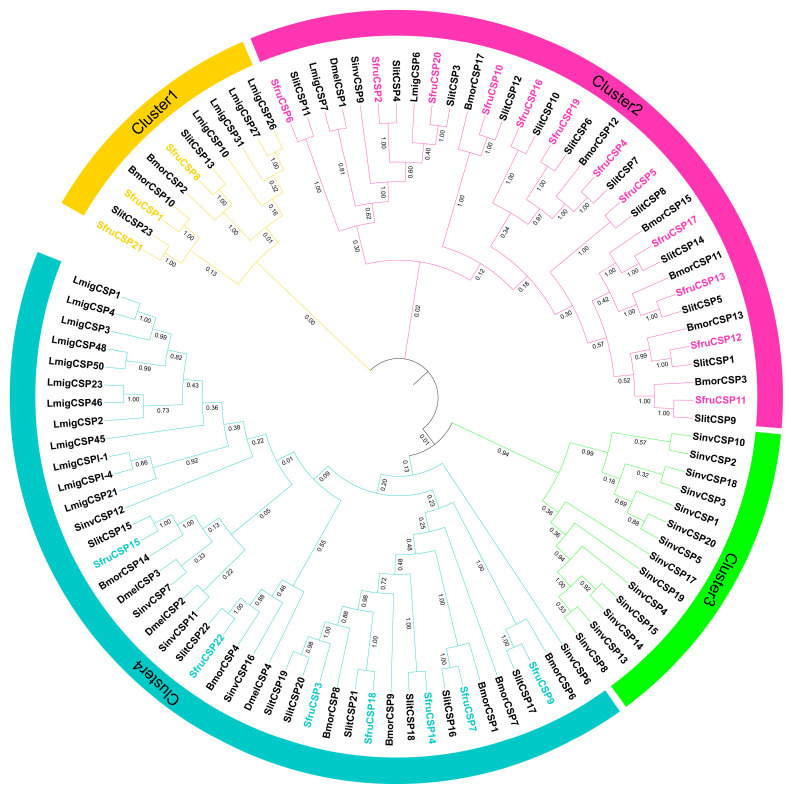
Phylogenetic analysis of *CSP* representatives from six species. The prefixes Sfru, Slit, Bmor, Lmig, Dmel, and Sinv denote *CSP* proteins from *S. frugiperda, S. litura, B. mori, L. migratoria, D. melanogaster,* and *S. invicta*, respectively. The SfruCSP genes are binned into various clusters based on the homology among *CSP* representatives.

**Figure 3 ijms-24-05595-f003:**
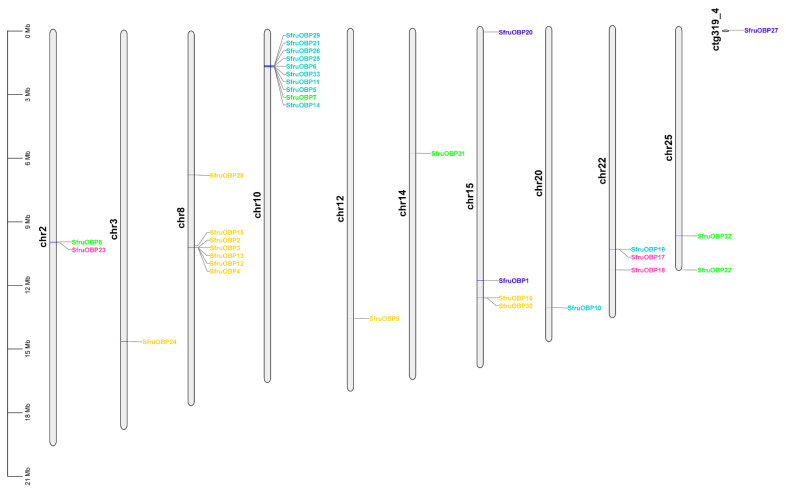
Chromosomal localization of the *SfruOBP* genes based on the genome data of *S. frugiperda*. Different colors of gene names indicate *OBPs* from specific Clusters as indicated in Figure 1: cyan, Cluster 1; pink, Cluster 2; bright green, Cluster 3; blue, Cluster 4; orange, Cluster 5.

**Figure 4 ijms-24-05595-f004:**
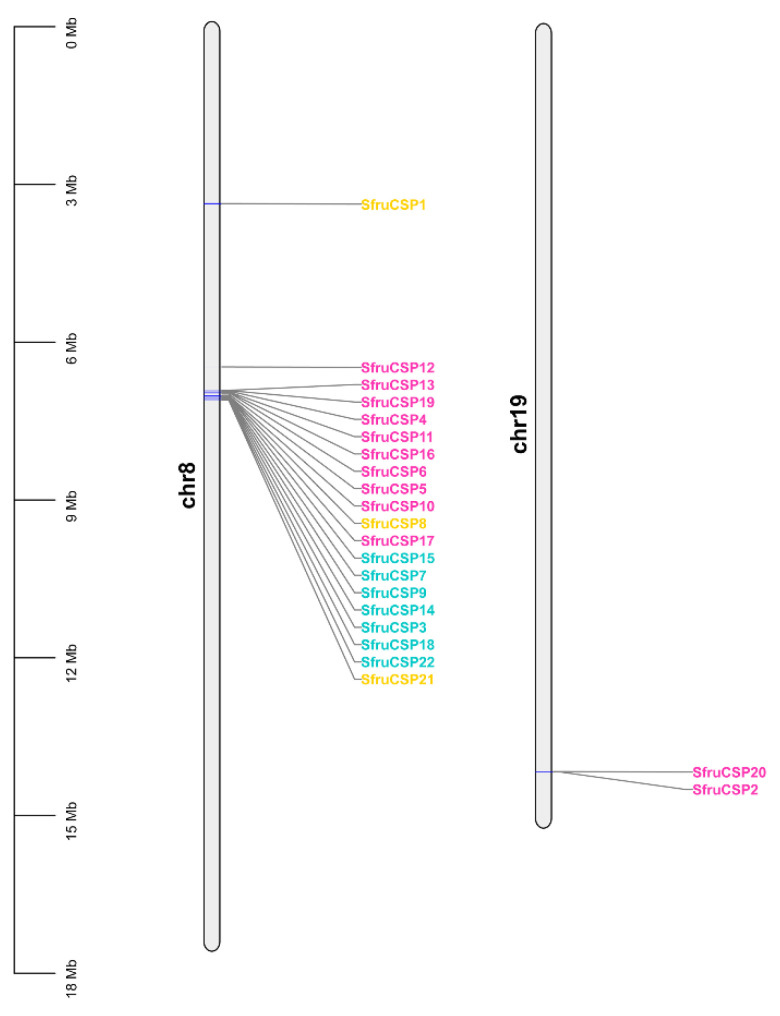
Chromosomal localization of the *SfruCSP* genes based on the genome data of *S. frugiperda*. Different colors of gene names indicate *CSPs* from specific clusters as indicated in Figure 2: orange, Cluster 1; pink, Cluster 2; bright green, Cluster 3; cyan, Cluster 4.

**Figure 5 ijms-24-05595-f005:**
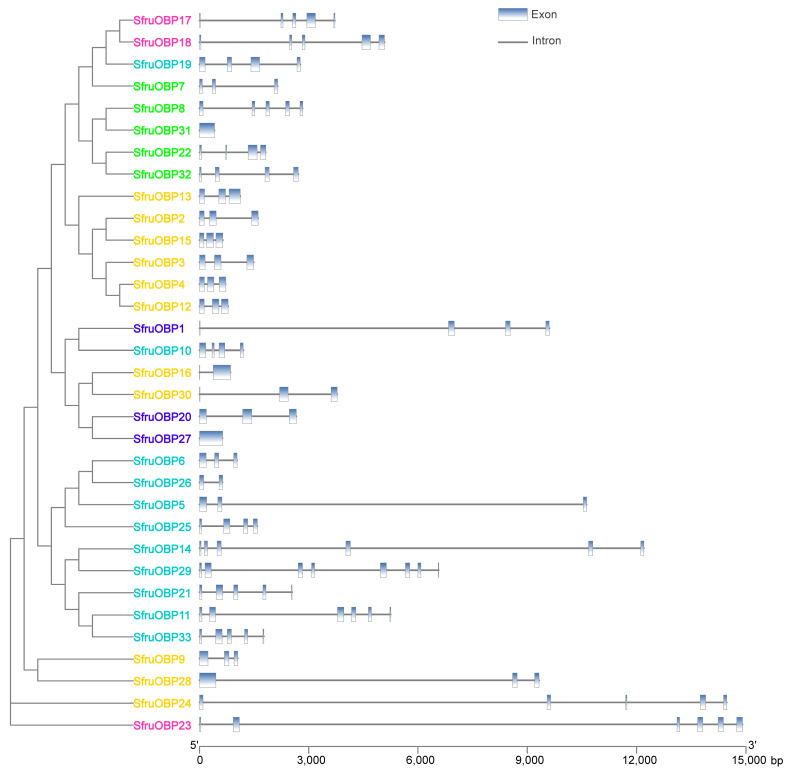
Exon–intron structures of *SfruOBP* genes. The blue boxes represent exons and the black lines indicate introns. The *x*-axis below scales the nucleotide sequence length (bp) of the gene. The full-length intron/exon composition of each gene is reported in Table 1. Different colors of the gene names indicate *OBPs* from specific Clusters as indicated in Figure 1: cyan, Cluster 1; pink, Cluster 2; bright green, Cluster 3; blue, Cluster 4; orange, Cluster 5.

**Figure 6 ijms-24-05595-f006:**
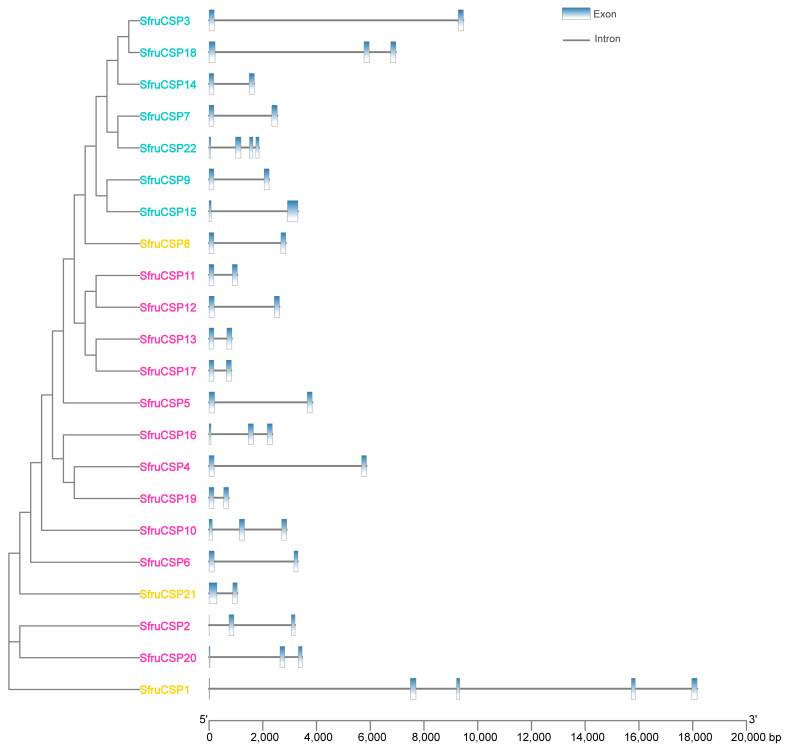
Exon–intron structures of *SfruCSP* genes. The blue boxes represent exons and the black lines indicate introns. The *x*-axis below scales the nucleotide sequence length (bp) of the gene. The full-length intron/exon composition of each gene is reported in Table 2. Different colors of the gene names indicate *CSPs* from specific clusters as indicated in Figure 2: orange, Cluster 1; pink, Cluster 2; bright green, Cluster 3; cyan, Cluster 4.

**Figure 7 ijms-24-05595-f007:**
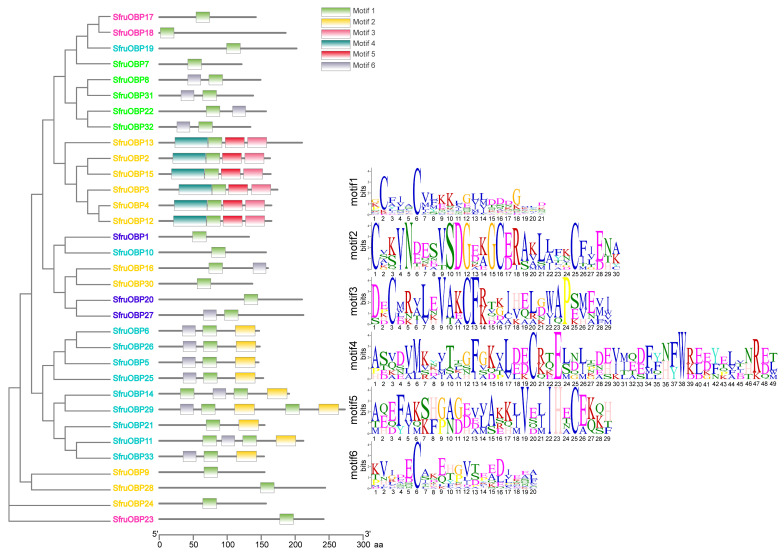
Motif distribution of *SfruOBPs*. Different motifs are represented by different colors, and the *x*-axis represents the length of the proteins. The SeqLogo of motifs is predicted with the MEME online tool. Different colors of gene names indicate *OBPs* from specific Clusters as indicated in Figure 1: cyan, Cluster 1; pink, Cluster 2; bright green, Cluster 3; blue, Cluster 4; orange, Cluster 5.

**Figure 8 ijms-24-05595-f008:**
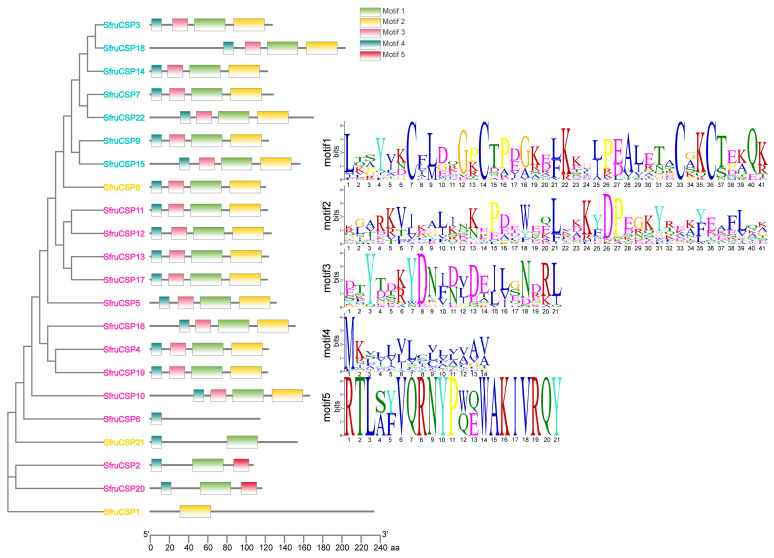
Motif distribution of *SfruCSPs*. Different motifs are represented by different colors, and the *x*-axis represents the length of the proteins. The SeqLogo of motifs is predicted with the MEME online tool. Different colors of gene names indicate *CSPs* from specific clusters as indicated in Figure 2: orange, Cluster 1; pink, Cluster 2; bright green, Cluster 3; cyan, Cluster 4.

**Figure 9 ijms-24-05595-f009:**
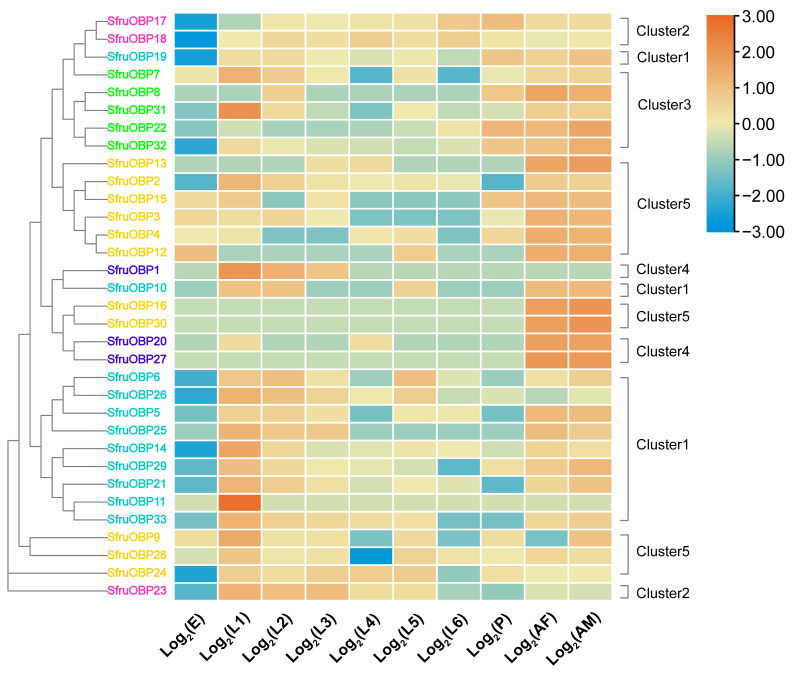
Expression heat map of *SfruOBPs* in eight developmental stages and across adult sexes (E: eggs; L: larvae; P: pupae; A: adult, where F: females and M: males). Expression values are scaled by row. The level of expression is indicated by different colors (right). Yellow represents positive expression, and the darker the color, the higher the expression level; blue represents negative expression, and the darker the color, the lower the expression level. For each sample we used three replicates. Different colors on gene names indicate *OBPs* from specific Clusters as indicated in Figure 1: cyan, Cluster 1; pink, Cluster 2; bright green, Cluster 3; blue, Cluster 4; orange, Cluster 5.

**Figure 10 ijms-24-05595-f010:**
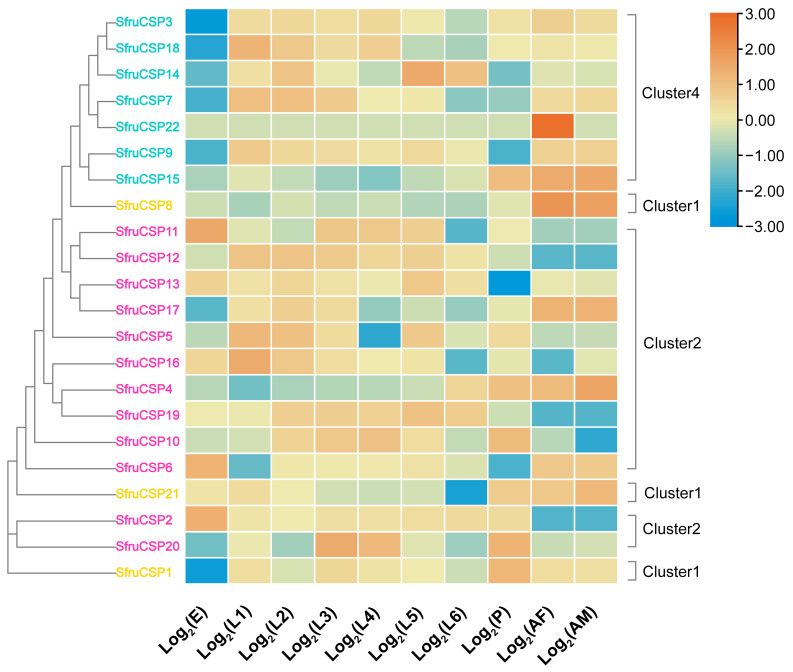
Expression heat map of *SfruCSPs* in eight developmental stages and across adult sexes (E: eggs; L: larvae; P: pupae; A: adult, where F: females and M: males). Expression values are scaled by row. The level of expression is indicated by different colors (right). Yellow represents positive expression, and the darker the color, the higher the expression level; blue represents negative expression, and the darker the color, the lower the expression level. For each sample we used three replicates. Different colors of gene names indicate *CSPs* from specific clusters as indicated in Figure 2: orange, Cluster 1; pink, Cluster 2; bright green, Cluster 3; cyan, Cluster 4.

**Figure 11 ijms-24-05595-f011:**
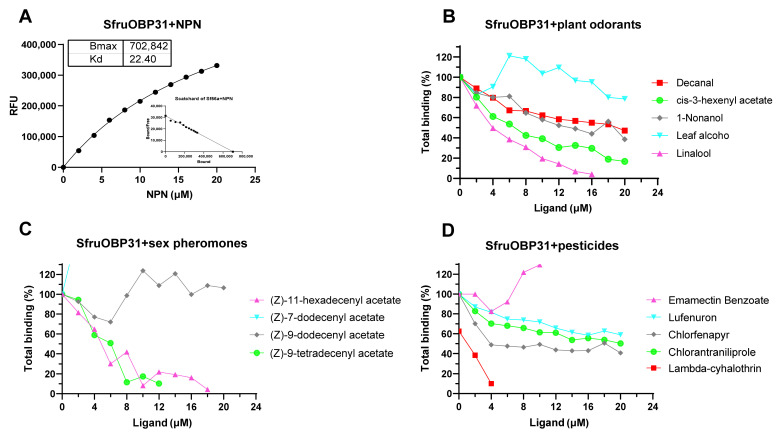
Binding assays of *SfruOBP*31 testing volatiles from host plants, pheromones, and pesticides. (**A**) Binding curves and dissociation constants of *SfruOBP*31 with 1-NPN. (**B**) Competitive binding curves of *SfruOBP*31 with five host volatiles. (**C**) Competitive binding curves of *SfruOBP*31 with four pheromones. (**D**) Competitive binding curves of *SfruOBP*31 with five pesticides.

**Table 1 ijms-24-05595-t001:** Physical and molecular properties of 33 *SfruOBPs* identified in the Fall armyworm (*S. frugiperda*). ND: not detected.

Gene Name	Genome ID	Chromosome No.	Length (bp)	Introns	Exons	Amino Acids (aa)	Molecular Weight (kDa)	Isoelectric Point	Signal Peptide (aa)
OBP1	Sfru136920	chr15	9606	2	3	131	14.90	9.01	ND
OBP2	Sfru112680	chr8	1607	2	3	162	18.23	5.31	1-21
OBP3	Sfru112690	chr8	1488	2	3	173	19.14	5.16	1-30
OBP4	Sfru112730	chr8	722	2	3	164	18.75	5.47	1-23
OBP5	Sfru220490	chr10	10,632	2	3	146	16.16	4.47	1-21
OBP6	Sfru220520	chr10	1035	2	3	147	15.73	5.42	1-21
OBP7	Sfru220480	chr10	2148	2	3	121	13.97	8.15	ND
OBP8	Sfru131490	chr2	2834	4	5	149	17.22	4.63	1-26
OBP9	Sfru078880	chr12	1052	2	3	155	18.22	4.88	ND
OBP10	Sfru072390	chr20	1203	3	4	118	12.69	8.72	1-20
OBP11	Sfru220500	chr10	5252	5	6	211	23.91	5.52	1-23
OBP12	Sfru112710	chr8	783	2	3	164	18.74	5.36	1-19
OBP13	Sfru112700	chr8	1121	2	3	209	23.92	8.07	1-22
OBP14	Sfru220470	chr10	12,205	5	6	191	21.37	5.84	ND
OBP15	Sfru112570	chr8	644	2	3	164	19.27	5.51	1-19
OBP16	Sfru136280	chr15	852	1	2	159	18.58	8.46	1-19
OBP17	Sfru036850	chr22	3719	4	5	141	15.70	6.86	ND
OBP18	Sfru153150	chr22	5083	4	5	135	14.44	8.96	ND
OBP19	Sfru036860	chr22	2766	3	4	202	22.75	5.12	1-19
OBP20	Sfru194150	chr15	2659	2	3	160	18.14	9.97	ND
OBP21	Sfru220550	chr10	2544	4	5	154	16.98	4.89	1-23
OBP22	Sfru221430	chr25	1817	3	4	156	16.93	7.51	1-22
OBP23	Sfru131450	chr2	14,905	5	6	241	27.63	5.82	ND
OBP24	Sfru032040	chr3	14,476	4	5	156	17.81	6.38	ND
OBP25	Sfru220530	chr10	1592	3	4	152	16.32	4.69	1-21
OBP26	Sfru220540	chr10	635	1	2	148	16.15	4.89	ND
OBP27	Sfru116380	ctg319_4	635	0	1	211	24.78	6.91	ND
OBP28	Sfru007560	chr8	9327	2	3	244	26.94	5.61	1-18
OBP29	Sfru220560	chr10	6574	7	8	272	30.59	5.16	1-20
OBP30	Sfru136290	chr15	3781	2	3	136	15.53	7.63	1-19
OBP31	Sfru155250	chr14	413	0	1	137	14.71	4.36	1-20
OBP32	Sfru139020	chr25	2713	3	4	133	15.09	8.88	1-16
OBP33	Sfru220510	chr10	1769	4	5	153	17.10	4.72	1-21

**Table 2 ijms-24-05595-t002:** Physical and molecular properties of 22 *SfruCSPs* identified in the Fall armyworm (*S. frugiperda*). ND: not detected.

Gene Name	Genome ID	Chromosome No.	Length (bp)	Introns	Exons	Amino Acids (aa)	Molecular Weight (kDa)	Isoelectric Point	Signal Peptide (aa)
CSP1	Sfru089520	chr8	18,161	4	5	233	25.59	10.44	ND
CSP2	Sfru158600	chr19	3201	2	3	107	11.94	9.49	1-22
CSP3	Sfru069670	chr8	9467	1	2	127	14.80	6.74	1-18
CSP4	Sfru007770	chr8	5864	1	2	123	13.57	5.23	1-18
CSP5	Sfru069550	chr8	3841	1	2	131	15.61	9.42	1-25
CSP6	Sfru007800	chr8	3306	1	2	114	12.92	5.04	1-16
CSP7	Sfru069640	chr8	2542	1	2	128	14.60	5.43	1-16
CSP8	Sfru069600	chr8	2859	1	2	120	13.77	5.86	1-16
CSP9	Sfru069650	chr8	2240	1	2	123	14.36	6.82	1-18
CSP10	Sfru069590	chr8	2891	2	3	166	18.59	9.57	ND
CSP11	Sfru007780	chr8	1057	1	2	122	13.82	8.95	1-16
CSP12	Sfru007430	chr8	2617	1	2	126	14.10	7.65	1-15
CSP13	Sfru007750	chr8	853	1	2	123	13.78	8.77	1-16
CSP14	Sfru069660	chr8	1679	1	2	122	14.20	6.12	1-16
CSP15	Sfru069630	chr8	3308	1	2	156	18.30	9.04	ND
CSP16	Sfru007790	chr8	2358	2	3	151	17.14	5.61	ND
CSP17	Sfru069620	chr8	828	1	2	122	14.03	5.84	1-17
CSP18	Sfru069680	chr8	6950	2	3	203	23.31	6.90	ND
CSP19	Sfru158610	chr19	3462	2	3	116	12.87	9.38	1-29
CSP20	Sfru069530	chr8	735	1	2	122	13.62	5.14	1-17
CSP21	Sfru069700	chr8	1054	1	2	153	17.12	8.86	ND
CSP22	Sfru069690	chr8	1857	3	4	170	19.22	4.93	ND

## Data Availability

Data are contained within the article or Appendix A.

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
