# Peer review of "Odorant-Binding Proteins and Chemosensory Proteins in Spodoptera frugiperda: From Genome-Wide Identification and Developmental Stage-Related Expression Analysis to the Perception of Host Plant Odors, Sex Pheromones, and Insecticides"

_ijms, 2023, doi:10.3390/ijms24065595_

Round 1

Reviewer 1 Report

The manuscript screened for genome-wide OBPs and CSPs of S. frugiperda, analyzed their gene expression patterns across the whole developmental stages, and the binding ability of SfOBP31 with some plant ordorants, pheromones and insecticides. The researches involved in this paper is simple, clear and easy to understand, but the innovation, depth and richness are not high. By analyzing the previously published genome and transcriptome data of S. frugiperda, many researchers may get the results in this paper. The low innovation and few molecular scientific results makes the content of the article not suitable for the journal. 

In addition, I have some specific comments.

1. The number of figures in this paper is too large, and the clarity of some pictures, such as figures 1, 3 and 7, need to be improved.

2. It can be seen from figure 9 that the expression of OBP11 rather than OBP31 is the highest in the first instar larvae. In addition, the olfactory of the first instar larvae is generally underdeveloped, and the food mainly depends on the parents to choose the oviposition site, so it is far-fetched to choose OBP by evaluating the expression level. In fact, there is no clear and reasonable explanation for why a single OBP31 in larvae is selected to analyze its binding ability with some plant ordorants, pheromones and pesticides, and the relevant conclusions are not of scientific significance.

3. Some previously published articles have screened and identified the OBPs and CSPs of S. frugiperda. Compared with these articles, what is the innovation of this article?

4. In this paper, several OBPs and CSPs without signal peptides were screened, and how to analyze the function of these proteins.

5. In the result 2.3, there are no relevant results to overview “gene structures”.

6. It is suggested that the correlation between phylogeny and function of genes arranged in clusters on chromosomes should be analyzed.

Reviewer 2 Report

The study authored by Chen Jia et al., investigated the evolutionary genomic clustering, chromosome localization, gene structural patterns, and expression pattern of OBP and CSP proteins. Furthermore, the binding affinity of plant odorants, sex pheromones, and pesticide's interaction with SfruOBP31. The objectives and importance of the research were clearly mentioned in the introduction and discussion part. However, the manuscript needs few clarification and typo error  modification in the revision.

Major comments

1. Abstract: The content expressed in the abstract well, but the rearrangement of sentence could provide good continuation for a reader. Hence, I suggest the authors to rewrite the abstract for easy understanding. I have herewith given a rephrase sentence suggestion for the authors.

"Spodoptera frugiperda is a worldwide generalist pest with remarkable adaptations to en vironments or stresses, including developmental stage-related behavioral and physiological adap tations, such as diverse feeding preference, mate seeking, and pesticide resistance. Insects’ odorant binding proteins (OBPs) and chemosensory proteins (CSPs) are essential for the chemical recogni tion during behavioral responses or other physiological processes. The genome-wide identification and the gene expression patterns of all these identified OBPs and CSPs across developmental stage related S. frugiperda have not been reported. We found 33 OBPs and 22 CSPs in the S. frugiperda genome. The majority of the Sfru OBP genes were most highly expressed in the adult male or female stages, while more SfruCSP genes were highly expressed in the larval or egg stages, indicating their function complementation. Here, we screened for genome-wide SfruOBPs and SfruCSPs and the gene expression patterns of SfruOBPs and SfruCSPs repertoires across the whole developmental stages, and sexes, were analyzed. The gene expression patterns of SfruOBPs and SfruCSPs reveal strong correlations with their respective phylogenic trees, indicating a correlation between function and evolution. In addition, we analyzed the chemical competitive binding of a widely expressed protein, SfruOBP31, to host plant odorants, sex pheromones, and  insecticides. Further ligands binding assay revealed a broad functional related binding spectra of SfruOBP31 to host plant odorants, sex pheromone and insecticides, suggesting its potential function in food, mate seeking, and pesticide resistance. These results provide guidance for future research on the development of behavioral regulators of S. frugiperda or other environmentally friendly pest control strategies."

2. Table 1. The 33 OBP and 22 CSP protein fasta sequence must be provided in the supplementary file, because it is difficult to verify the Gene ID (for ex. Sfru136920 in the NCBI). Is that valid Gene ID? the same question on the table 2 CSPs gene ID? Provide correct gene id for the proteins.

3. Figure 1 and 3 are not clearly visible, bootstrap values and letters respectively. Furthermore, Figure 7 is not clear, change the figure accordingly.

4. Why the authors have choosed the SfruOBP31 for the binding affinity analysis? SfruOBP31 is not showing good expression on L3/L4/L6/P and E stages.

5. Line 369: The authors informed about that this is the first study of identifying OBP/CSPs genome in Sfru. but line 371-372 (Ref.45), they have compared to the previous OBP/CSPs in the same genome of Sfru study? it is contrary to the readers.

6. Line 391-393: Verify the sentence. Not all the 5 clusters of SfruOBPs are similar gene structure and expression pattern. Cluster 1/2/3/4 are not similar pattern in exon-intron structure within the cluster of SfruOBP, itself, except cluster 5.

7. Line 437: higher expression in larvae L1 stage but not in the AF/AM?

8. Line 445-447: is that correct? A suggestion to authors to verify the sentence with the result obtained, There is no similar gene structure and expression pattern between the cluster and within  the clusters of SfruOBPs except cluster5. The cluster five SfruOBP only has similar exon-intron structure but not the similar expression heat map pattern.

9. Line 495: is that correct reference cited? If yes, mention the female sex pheromone in the text.

10. Line 525: "high resolution genome" seems like experimentally evaluated genes with the visualization of structure with good resolution. The authors have not validated by crystallographic all the proteins. Change the word in all over the manuscript to in-sight/indepth view of genome" may be.

Minor comments

1. Line 22: Abstract- Remove bold in "Spodoptera"
2. Line 67: What are the alternative physiological functions in the insect?
3. Line 76-79: What is the classical OBP? in which species or all the mammals?
4. Line 85-90: Where the CSPs are expressed?
5. Line 93-97: indicate a life span of the insect?
6. Line 126-127: How many OBPs are classical OBP group? The authors mentioned, "Twenty-two OBPs belong to the classical OBP group (SfruOBP1-14, 16, 19-21, 24, 26-28, 30-33)" but is it 26 genes mentioned in the SfruOBP1-14, 16, 19-21, 24, 26-28, 30-33?
7. Line 136: indicate the conserved Cysteine number in the supplementary table 1.
8. Line 143: Error in OBP code verify. Is it OBP29 not SfruOBP7?
9. Line 146: OBP3 showed 5.16 Kda? Is it correct?
10. Line 144: What is the mean average length of the amino acids?
11. Line 152: sure about 15-30 amino acids? 1-26 signal peptide max in both OBP and CSP.
12. Table 1: OBP3 had 173 exons? Verify the whole numbers, there are several errors in the line.
13. Figure 2: SiCSP6 present in the cluster 4, is it outgroup?  Why it is not aligned with cluster 3?
14. Figure 5: indicate the intron blact line legend inside the figure.
15. Line 235: numbers indicating nucleotide?  bp? length of the gene?
16. Line 252: The authors indicated the figure.8 but not in the result part, rather it is  in the discussion? why. Replace to the result part.
17. Line 266: Explain what is instars?
18. Line 363: modify to "The soluble chemosensory proteins"
19. Line 378: modify the error to "SfruOBPs"

Round 2

Reviewer 1 Report

Most of the comments have been revised.

It is suggested that the authors should detect the chemical competitive binding of other OBPs which is different from the expression pattern of OBP31 to the same chemical substances  in order to make the conclusion more convincing.

Reviewer 2 Report

Appreciation to all the authors for the modification and changes have incorporated in the revised version according to the reviewer comments. Since, I have few minor comments for the improvement of the article.

Minor comments
1. Table-1, What is mean the genome ID ? Where is it obtained ? from which database, if NCBI, indicate the accession number and include it in the table or supplementary files. It is difficult to find the sequence information in the database. Did the authors have submitted these sequences to the database?

2. Figure 5: Error in figure internal legend, "intron" not intro correct it accordingly.
